# Mycotoxin Metabolism by Edible Insects

**DOI:** 10.3390/toxins14030217

**Published:** 2022-03-17

**Authors:** Natasha Marie Evans, Suqin Shao

**Affiliations:** Guelph Research and Development Centre, Agriculture and Agri-Food Canada, Guelph, ON N1G 5C9, Canada; natasha.evans@uwaterloo.ca

**Keywords:** mycotoxins, edible insects, aflatoxins, fumonisins, zearalenones, deoxynivalenol, ochratoxins, mycotoxin metabolism

## Abstract

Mycotoxins are a group of toxic secondary metabolites produced in the food chain by fungi through the infection of crops both before and after harvest. Mycotoxins are one of the most important food safety concerns due to their severe poisonous and carcinogenic effects on humans and animals upon ingestion. In the last decade, insects have received wide attention as a highly nutritious, efficient and sustainable source of animal-derived protein and caloric energy for feed and food purposes. Many insects have been used to convert food waste into animal feed. As food waste might contain mycotoxins, research has been conducted on the metabolism and detoxification of mycotoxins by edible insects. The mycotoxins that have been studied include aflatoxins, fumonisins, zearalenone (ZEN), vomitoxin or deoxynivalenol (DON), and ochratoxins (OTAs). Aflatoxin metabolism is proved through the production of hydroxylated metabolites by NADPH-dependent reductases and hydroxylases by different insects. ZEN can be metabolized into α- and β-zearalenol. Three DON metabolites, 3-, 15-acetyl-DON, and DON-3-glucoside, have been identified in the insect DON metabolites. Unfortunately, the resulting metabolites, involved enzymes, and detoxification mechanisms of OTAs and fumonisins within insects have yet to be identified. Previous studies have been focused on the insect tolerance to mycotoxins and the produced metabolites; further research needs to be conducted to understand the exact enzymes and pathways that are involved.

## 1. Introduction

Mycotoxins are secondary metabolites that are commonly produced by certain species of fungi, such as mould [1]. Hundreds of different types of mycotoxins, such as deoxynivalenol (DON), aflatoxin B1 (AFB1), and zearalenone (ZEN), have been identified in nature and habitually contaminate crops that are intended for commercial purposes [2]. Additional mould growth after harvesting can arise from the warm and moist environmental conditions that are often present during the processing and storage of food crops [1]. When exposed to mycotoxins through ingestion, both humans and animals can experience severe health consequences including poisoning and organ damage, as well as carcinogenic and mutagenic effects [3]. Therefore, most contaminated crops cannot be used for commercial purposes and are habitually sorted out for disposal [4,5]. Mycotoxins have been identified as the major food contaminants that affect global food safety and food security, especially in underdeveloped countries.

However, the global population growth has driven a significant increase in the demand for food, especially for meat. Increasing traditional food production, including meat, has been challenging because of the environmental impact. Food industries are increasingly shifting their focus towards the use of alternative protein sources, such as insects. The consumption of insects has already been widely accepted in some countries and there is rapidly growing interest in raising insects as animal feed around the world.

Although the growth and development of insects are also affected by mycotoxins, they are often able to withstand considerably high concentrations of these compounds [2]. To understand the metabolic capabilities and potential toxic effects to the organisms, studies have been conducted in which edible insects were exposed to artificially elevated and naturally occurring levels of mycotoxins [6]. The resulting concentrations of mycotoxins and subsequent metabolites within the insects after short or prolonged periods of exposure were analyzed to further understand how these organisms manage the toxins [7]. With the aim to provide the general state of scientific research in this area and identify gaps for future research, the scientific literature on insects and mycotoxins published in the last 20 years was searched in Agricola databases, CAB databases and Scopus. The literature on edible insects was first summarized, and then the metabolism of insects was summarized based on the group of mycotoxins including aflatoxins, fumonisins, zearalenones, vomitoxin or deoxynivalenol, and ochratoxins. Future directions and recommendations on this topic are provided in the Conclusions.

## 2. Edible Insects

### 2.1. Insects as an Alternative Protein Source for Humans

As populations continue to rise, the global demand for meat has increased exponentially, putting numerous strains on the environment [8]. Since the production of meat requires vast amounts of land and resources, food industries are increasingly shifting their focus towards the use of alternative protein sources, such as insects, for food products [8]. The consumption of insects has already been widely accepted and implemented in countries located within Central and West Africa, as well as South East Asia [8]. However, efforts to shift the food industry towards insect-based proteins in Western societies has proven to be more difficult, since the consumption of insects is often perceived with repulsion and skepticism [8].

Insects make ideal livestock as they can thrive within highly dense populations in small spaces, can grow quickly with high survival rates, and are highly reproductive [9]. In addition, the production of insects is environmentally efficient as it requires far fewer resources and results in a significant reduction in emitted greenhouse gases compared with traditional livestock, such as cattle [9]. The rearing of insects can also be incredibly cost efficient as they are capable of bio-converting livestock manure into food [10]. Insects are also a rich source of protein, with some species being able to provide up to 48% of edible protein content [11]. This is comparable to the fresh-weight protein content of animals, such as fish and cattle [11]. Not only do insects help to fulfill nutritional requirements, but they also have the ability to provide numerous health benefits for humans [12]. In Korea, silkworms have been used for diabetic medicines due to their blood-glucose lowering properties [12]. Additionally, certain species of insects, such as termites, have been found to have immunostimulatory effects on humans [12].

However, the nutritional content and health benefits of insects vary from species to species [13]. For instance, some groups of insects are not advised to be used as meat substitutes due to their elevated sodium and saturated fatty acids content, which can increase the risk of heart diseases [13]. On the other hand, several species of insects that can be found in areas of food insecurity and micronutrient deficiency contain high quantities of micronutrients, which can be used to fight malnutrition in areas where food can be scarce [13]. 

Despite the abundant benefits of shifting towards an insect-based protein industry, there are instances where food safety can be compromised [12]. For example, the consumption of insects may induce allergic reactions within the consumer [12]. Although relatively few studies have been conducted on allergic reactions caused by the ingestion of insects, it has been found that certain species of edible insects have allergy cross-reactive proteins with crustaceans, which can pose a danger to certain people [12]. On the other hand, the insects may act as a vector for pathogenic bacteria [12], thus affecting the intestinal microbiota of the consumer [8]. Finally, the toxins produced by certain insects can accumulate within the body of the consumer over time and potentially lead to detrimental effects in the future [12].

### 2.2. Insects as Animal Feed vs. Other Animal Feed

As the global demand for meat continues to rise at an exponential rate, unsustainable quantities of feed are required for livestock, leading to increased stress on limited resources [14]. Thus, the meat industry is turning towards insects as a source of protein for animal feed due to low production costs and high nutritional properties [15]. This can have positive physiological effects on livestock such as poultry and swine, as well as aquaculture [15]. A study conducted by Dabbou et al. [14] monitored the effects of partially substituting traditional protein sources of broiler chicken feed, such as soybeans, with black soldier fly (*Hermetia illucens*) larvae. *H. illucens* fulfill the nutritional requirements of chicken as they contain a high protein content of 37–63%, as well as a superior amino acid profile compared with soybeans [14]. Over a 35-day study, the chicks were provided with feed that contained 0, 5, 10 and 15% of *H. illucens* larva meal [14]. From the results, it was determined that the inclusion of 10% *H. illucens* greatly improved the growth of chicks [14]. However, when the percentage was raised to 15%, a decrease in feed conversion ratio was observed [14]. Similarly, it was observed that high percentages of *H. illucens* caused changed intestinal morphology within the broilers [14]. The intestines of chicks that were fed 15% *H. illucens* exhibited deeper crypts, shorter villi, and reduced villus height-to-crypt-depth ratios [14]. Despite the physiological effects, the health of broilers was not impacted by the presence of *H. illucens* [14]. This was observed as the heterophils-to-lymphocytes ratio remained similar for each group of chicks [14]. Therefore, the overall data collected from the study suggest that the protein source of conventional animal feed should only be partially substituted with insects to reduce the negative effects on livestock [16].

However, the nutritional and physiological effects on livestock change drastically based on the type of insect added to the feed [16]. For example, while *H. illucens* larvae are a source of high protein and vitamin content [2,17], other insects such as grasshoppers lack a sufficient amino acid profile and contain a crude protein fraction that is difficult to digest [16]. Therefore, the consumption of grasshoppers is associated with a decrease in average daily gain and thus negatively impacts the growth of poultry [16]. Despite the physiological changes caused by the partial incorporation of insects in animal feed, the overall flavour and texture of the final food products are not compromised in any way [15].

## 3. Metabolism of Mycotoxins by Edible Insects

Insects are naturally exposed to mycotoxins in the environment; therefore, they contain detoxification mechanisms which allow for both the breakdown and excretion of ingested toxins [2]. Like mammals, insects possess phase I and II metabolic enzymes, such as glycosyltransferase and cytochrome P450 monooxygenase (P450), that are essential for mycotoxin reduction [7]. A summary of publications that have focused on mycotoxin detoxification by edible insects is outlined in Table 1. Although many studies have investigated insect tolerance to mycotoxins and the produced metabolites, further research needs to be conducted to understand the exact enzymes and pathways that are involved.

### 3.1. Aflatoxins

Aflatoxins are a group of mycotoxins, produced by fungi belonging to the *Aspergillus* species, that commonly contaminate cereal foods such as peanuts and corn [27]. These toxins have been found to not only affect field crops but also animal-derived food products, including dairy products, when livestock consume contaminated feed [27]. Of the different forms of this mycotoxin, AFB1 is one of the most dangerous metabolites due to its immunosuppressant and carcinogenic properties towards numerous animal species and humans [28]. For instance, prolonged exposure to aflatoxin contaminated feed produces severe health effects in cattle and swine, such as respiratory illnesses and liver damage, respectively [29]. AFB1 can be hydroxylated into a less toxic metabolite, aflatoxin M1 (AFM1); however, this degradation product is still of concern to consumers as it poses similar health impacts to its parent compound [28].

Certain insect species can withstand high concentrations of aflatoxins, as they populate environments with high levels of fungal growth in their food sources [30]. For instance, investigations conducted by Niu [19] analyzed the P450-mediated detoxification of AFB1 in honeybees (*Apis mellifera*, human food [20] and corn earworms (*Helicoverpa zea*), feed for chicken [22]), as described in Table 1. Evidence of aflatoxin metabolism within these organisms is apparent through the production of hydroxylated metabolites, such as AFM1, aflatoxicol (AFL), aflatoxin P1 (AFP1) and aflatoxin B2a (AFB2a), by NADPH-dependent reductases and hydroxylases [19,21,30]. Further confirmation of AFB1 tolerance within these organisms can be observed through a lack of metabolic pathways that convert AFB1 into the more toxic compound, AFB1-8,9-epoxide (AFBO) [19,30]. This epoxide compound is highly genotoxic and carcinogenic, due to its ability to form DNA adducts within host cells [30]. Moreover, a publication by Bosch et al. [23] analyzed the effects of AFB1 and its metabolite AFM1 on yellow mealworms (*Tenebrio molitor*) and *H. illucens*, as shown in Table 1. Through this study, the insect larvae were reared on poultry feed spiked with 0.1–0.5 mg/kg of AFB1 over a period of 10 days for *H. illucens* and until the first pupa hatched for *T. molitor* [23]. From the results, it was found that these insect species can withstand AFB1 concentrations of up to 0.415 mg/kg, with minimal impacts on their development or survival [23]. In addition, the gut contents of *T. molitor* were revealed to play an important role in AFB1 metabolism [23]. For example, larvae that were given non-contaminated feed for an additional 2 days after exposure exhibited lower AFB1 concentrations compared with larvae that were analyzed directly after consumption of the contaminated feed [23]. Both insect species were also found to contain AFB1 and AFM1 levels that were either below the legal limit of 0.02 mg/kg that has been set for European feed material or under analytical detection levels [23]. Although these studies aid in understanding the metabolic mechanisms that certain edible insects undergo when exposed to aflatoxins, there are many additional mycotoxins that can affect the quality of insects and their chemical compositions [23].

### 3.2. Fumonisins

Fumonisins are carcinogenic metabolites produced by the fungus, *Fusarium moniliforme*, and like aflatoxins, these mycotoxins are of global concern due to their frequent presence in corn crops [31]. These toxins, which include fumonisin A1 (FA1), fumonisin A2 (FA2), fumonisin B1 (FB1), fumonisin B2 (FB2), and fumonisin B3 (FB3) [26], are believed to be linked to esophageal cancer in humans who habitually consume contaminated corn [31]. Within this mycotoxin group, FB1 is one of the most concerning metabolites for mammals, due to its cytotoxic properties, which have been linked to the development of liver damage and cancer [26]. For instance, it was noted that its consumption affects the overall growth and internal development of chickens [26].

Evidence of fumonisin regulation by insects was observed in a study where *T. molitor* larvae were injected with a single dose of 0.25–25 ng/µL FB1, as shown in Table 1 [26]. Upon injection, the larvae’s respiratory metabolism was initially compromised, as they exhibited reduced levels of CO_2_ production [26]. However, the effects of mycotoxin regulation were detected a month after injection, as the CO_2_ levels returned to pre-exposure conditions for each concentration tested [26]. Thus, it was hypothesized by Abado-Becognee et al. [26] that *T. molitor* tolerance to FB1 can arise from immobilization within fat storages, direct excretion, or mycotoxin breakdown. Although *T. molitor* can recover from single exposures to FB1, it was found that larvae fed grains spiked with 50-450 mg/kg of FB1 over a period of 63 days experienced negative impacts on their growth and development [26]. For instance, the larvae that consumed the highest FB1 concentrations were slowly affected by the chronic toxin exposure through reduced growth, delayed moulting, and decreased CO_2_ production, as well as a decline in feed consumption compared with the control larvae [26]. Therefore, it appears that although *T. molitor* can reverse the metabolic effects of a single FB1 injection, chronic exposure to high concentrations has an impact on the overall health and nutritional value of these edible insects [26].

### 3.3. Zearalenones

Mycotoxins belonging to the zearalenone (ZEN) family are major cereal crop and animal feed contaminants that are also naturally produced by fungi of the *Fusarium* genus [32]. ZEN compounds have been linked to reproductive disorders in mammals due to their similarity to the female steroid hormone, 17β-estradiol [32,33,34]. For example, these mycotoxins may interrupt vital oestrogenic responses or hinder steroid secretion [32,34]. Within this mycotoxin family, it was found that α-zearalenol (ZEL) is the most potent metabolite due to its elevated oestrogenic activity [7,35]. Hence, these mycotoxins are of particular concern within the livestock industry as they can have oestrogenic effects on swine and cattle, such as hyperoestrogenism [32,33]. Unfortunately, ZEN compounds are difficult to treat as they can withstand temperatures of up to 160 °C; therefore, alternative detoxification strategies must be sought out [32,33].

When *T. molitor* larvae are exposed to high levels of ZEN, it can disrupt the function of their hormone ecdysone, which will in turn affect their development and moulting [7]. However, a study by Niermans et al. [7] found that *T. molitor* larvae are resilient towards wheat crops that contain ZEN concentrations as high as 2 mg/kg, as both the mycotoxin and its metabolites, α- and β-ZEL, were almost completely excreted from its system after exposure, as outlined in Table 1 [7]. Within the investigation, the larvae were fed wheat contaminated with 0.5–2 mg/kg of ZEN for a period of either 4 or 8 weeks [7]. When the larvae were analyzed by HPLC-MS, the chromatograms were not able to display any traceable levels of the mycotoxin [7], whereas the analyzed excreta contained ZEN, α- and β-ZEL concentrations that matched the initial mycotoxin concentrations present in the feed [7]. These results agree with a study conducted by Camenzuli et al. [2], where *H. illucens* and lesser mealworms (*Alphitobius diaperinus*) were given feed artificially spiked with 0.5–13 mg/kg of ZEN over a period of 10 days and 14 days, respectively. After exposure, the insects were given non-contaminated feed for an additional 2 days to determine the mycotoxin accumulation within the larvae’s body rather than the gut after ingestion [2]. From the results, the survival and growth of both insect species were not significantly impacted by any of the ZEN concentrations tested [2]. *H. illucens* larvae were found to metabolize over half of the ZEN consumed into α- and β-ZEL metabolites and excreted all these toxins when reared on spiked feed as well as when cleansed on uncontaminated feed [2]. On the other hand, *A. diaperinus* displayed minimal ZEN catabolism and were instead found to excrete almost all of the toxic compounds when reared on spiked feed [2]. Therefore, the different insect species were shown to conduct different detoxification strategies and neither species were severely affected by any of the ZEN concentrations tested [2].

### 3.4. Vomitoxins

Like fumonisins and zearalenone, vomitoxins are produced by fungi that belong to the *Fusarium* family and are a common grain contaminant [36,37]. Vomitoxin, also known as deoxynivalenol (DON), is a trichothecene that compromises the immune system and protein synthesis [37], and induces oxidative stress within animal cells upon ingestion [24]. In addition, exposure to DON may have potential impacts on the nervous system and feeding behaviour [37]. DON contamination is difficult to treat due to its stability; therefore, its presence in animal feed is of concern due to its influence on livestock productivity [36,37]. For instance, while cattle and poultry can handle DON exposure, horses and swine have been reported to refuse the consumption of contaminated feed [36].

A study by Van Broekhoven et al. [25] investigated *T. molitor*’s tolerance to DON through its breakdown and excretion mechanisms (Table 1). Within this study, T. molitor larvae were reared on wheat flour that was naturally or artificially contaminated with 4.9 and 8 mg/kg of DON, respectively, over the course of 2 weeks [25]. Not only did each of these DON concentrations have little effect on the overall growth and survival of the larvae, but the mycotoxin also did not accumulate within the organisms [25]. Analysis of the faeces revealed that the larvae excreted 14–40% of the DON that was initially present in the feed, but further information regarding the DON metabolites produced or the detoxification pathway could not be determined [25]. In addition, an investigation by Ochoa Sanabria et al. [4] investigated the effects of DON on *T. molitor* when given wheat contaminated with DON concentrations ranging from 2 to 12 mg/kg over the course of 30 days (Table 1). Within this study, although there were no lasting effects on the insect’s survival or growth, the larval bodies were found to contain negligible concentrations of DON and the metabolite 3-acetyl-DON that were on par with the levels detected in the control larvae [4]. The *T. molitor* larvae were found to only excrete 6–15% of unmetabolized DON, and instead excreted higher levels of 3-acetyl-DON, highlighting the organism’s metabolic detoxification capabilities [4]. A study by Janković-Tomanić et al. [24] further expanded on these results by increasing *T. molitor* larvae exposure through wheat contaminated with 4.9–25 mg/kg of DON over a period of 2 weeks (Table 1). Once again, the survival of *T. molitor* was not affected by the exposure; however, the larvae exhibited reduced locomotion, protein synthesis and body weight when exposed to the highest DON concentrations [24]. Through this study, it was also found that the activity of the enzymes glutathione-S-transferase (GST) and superoxide dismutase (SOD) increased proportionally to the DON exposure, in response to the resulting oxidative stress and to the detoxification of the newly formed superoxide anion radicals, respectively [24]. 

Furthermore, a study by Gulsunoglu et al. [5] investigated the effects of DON on *H. illucens* when fed grains contaminated with 0.63 mg/kg over the course of 12 days (Table 1). Like the studies with *T. molitor*, it was found that the bodies of *H. illucens* retained low concentrations of DON that were not of safety concern to consumers [5]. Likewise, in the study by Camenzuli et al. [2], *H. illucens* and *A. diaperinus* were given feed with DON concentrations that ranged from 5 to 125 mg/kg for 10 and 14 days, respectively (Table 1). From the results, the three DON metabolites tested, 3-, 15-acetyl-DON, and DON-3-glucoside, were below quantification limits for both species [2]. It was also discovered that 39–80% of the DON in the spiked feed was excreted from the *H. illucens* larvae while exposed to the toxin, as well as during the 2 days while reared on uncontaminated feed [2]. On the other hand, 80–96% of the original DON content was excreted from the *A. diaperinus* larvae while reared on contaminated feed [2]. Additionally, both studies once again observed that there were little to no effects on the growth, survival, and performance of the larvae [2,5].

### 3.5. Ochratoxins

Ochratoxins are a group of mycotoxins produced by fungi belonging the *Aspergillus* and Penicillium genera that commonly contaminate cereal crops due to the poor storage and handling of food products [38]. Within this mycotoxin family, ochratoxin A (OTA) is the most abundant and toxic form to mammals [38,39]. For instance, the toxin can affect the immune system [21], or target the kidneys and cause renal problems such as cancer or nephropathy in humans and animals alike [38]. Like the other naturally occurring mycotoxins, ochratoxins are chemically stable, and so can be difficult and costly to remove from contaminated feed [38]. Therefore, ochratoxin contamination in livestock feed is a recurring problem, thus leading to a decline in farm animal health and performance [40].

Within the studies conducted by Niu et al. [21] and Niu et al. [18], it was shown that both *A. mellifera* and *A. transitella* can tolerate high levels of ochratoxin A (OTA). This was demonstrated when *A. transitella* larvae were given feed contaminated with 1–50 mg/kg of OTA over a period of 14 days [21]. It was shown that *A. transitella* is relatively unaffected by OTA exposure and only experienced minimal developmental effects after ingesting feed with the highest contaminant levels [21]. Similarly, *A. mellifera* larvae were given candy contaminated with 1–80 mg/kg of OTA for a period of 72 h [18]. It was found that concentrations of 1 mg/kg or lower had no toxic effects on *A. mellifera*; however, the organisms were found to die within 3 days after the chronic consumption of candy with OTA concentrations higher than 5 mg/kg [18]. Likewise, the publication by Camenzuli et al. [2] discussed the OTA tolerance of *H. illucens* and *A. diaperinus* when exposed to concentrations ranging from 0.1 to 2.5 mg/kg for 10 and 14 days, respectively (Table 1). It was found that approximately 100% of the OTA was excreted from *A. diaperinus* when reared on both spiked and non-contaminated feed [2]. However, only about 50% of the original OTA was accounted for in the *H. illucens* larvae and excreta [2]. Unfortunately, the resulting metabolites, involved enzymes, and detoxification mechanisms of OTA within insects have yet to be identified [2,18,19,21].

### 3.6. Mycotoxin Mixture

In addition to understanding how edible insects react to individual mycotoxins, researchers have proceeded to analyze how these organisms handle exposure to a mixture of mycotoxins as they often co-exist in nature [2]. The study by Camenzuli et al. [2] investigated the tolerance of *H. illucens* and *A. diaperinus* towards feed that was artificially contaminated by a mixture of AFB1, ZEN, DON, and OTA mycotoxins with concentrations that exceed the maximum limits in European feed (0.02, 0.5, 5, 0.1 mg/kg, respectively) by 1, 10, and 25 times. Through this study, it was discovered that the insect’s response to a mixture of mycotoxins was no different than when exposed to feed contaminated with a single type of mycotoxin [2]. Hence, the insect’s ability to metabolize each individual mycotoxin was not affected by the existence or concentration of the other consumed mycotoxins [2]. Both insect species were able to mitigate the accumulation of all four mycotoxins through pathways including metabolic degradation in the gut or rapid excretion [2]. Exposure to the mixture of mycotoxins was also found to have little to no effect on the survival and development of the larvae [2]. However, *H. illucens* and *A. diaperinus* were found to metabolize the mycotoxin mixtures at different rates [2]. For instance, when exposed to mycotoxin concentrations that were 25 times the maximum European limit, *H. illucens* larvae were able to fully remove AFB1 from their system, but still contained concentrations of ZEN, DON, and OTA that were above the maximum European limits [2]. On the other hand, *A. diaperinus* larvae exhibited quantities below the maximum limits for each concentration tested [2]. A similar study was conducted by Leni et al. [41], where *A. diaperinus* and *H. illucens* larvae were given feed consisting of cereal and vegetable waste that was naturally contaminated with the mycotoxins ZEN, DON, FB1, and FB2. This study is a better reflection of the kind of situations that insects would potentially face in nature [41]. From the results, it was shown that the mycotoxins present within each insect species were either below detection levels or significantly reduced compared with original concentrations [41]. Therefore, these studies reveal potential opportunities to help reduce crops destined for waste by rearing insects on feed contaminated with a mixture of mycotoxins that far exceed the maximum safe concentration limits [2].

## 4. Insects That Are Fed with Mycotoxin Contaminated Grains as Animal Feed or Human Food

Researchers have investigated the possibility of using mycotoxin-contaminated crops as feed for reared insects due to their metabolic capabilities [4]. Each year, the agriculture industry experiences millions of dollars in losses due to mycotoxin contamination; therefore, using infected crops for insect feed can prove to be highly sustainable [4]. For example, grain crops infested with the fungal disease fusarium head blight contain high levels of DON, and cannot be sold in commercial markets due to their toxicity to consumers [5]. Thus, feeding insects mycotoxin-contaminated grains is an appealing alternative method compared with the use of traditional and time-consuming detoxification techniques [5], such as oxidation and irradiation, to recover these contaminated crops [42]. However, it is important to understand how mycotoxins ingested by insects may affect consumers [43]. The toxins accumulated within reared insects may transfer to humans or animals upon digestion or affect the growth and resulting nutrition of the insects [43].

In a study conducted by Guo et al. [44], it was discovered that *T. molitor* larvae prefer wheat kernels that have been artificially contaminated with the mycotoxin producing *Fusarium* species over non-contaminated kernels. Moreover, Ochoa Sanabria et al. [4] determined that using DON-contaminated wheat as feed had little to no effect on the production, survival or even the nutritional value of *T. molitor* larvae. Interestingly, it was also found that *T. molitor* larvae gained more weight when reared on feed containing 0.204 mg/kg of AFB1 [23], or natural ZEN contamination levels of 900 µg/kg [7], as opposed to uncontaminated feed. Furthermore, the chemical food safety of certain species of edible insects is not compromised by exposure to mycotoxins [2]. For instance, AFB1 and DON do not accumulate within *A. diaperinus* or *H. illucens* due to their metabolic detoxification pathways [2]. However, further studies must be conducted to understand the impacts of the degradation products on reared insects and the potential dangers to consumers [2]. For instance, *T. molitor* metabolizes the mycotoxin zearalenone (ZEN) into α-, β-zearalenol (ZEL) and almost completely excretes these compounds from its system [7]. However, the metabolite α-ZEL is a highly toxic oestrogen, and therefore although there are only trace amounts of the compound present, the toxicity may nevertheless have an influence on the organism’s reproductive system and on those who consume it [2,7]. Therefore, additional research must also be carried out for the different types of mycotoxins and edible insects, as every species responds uniquely to the various fungal toxins present in nature [43].

## 5. Conclusions

Most previous works have focused on insect tolerance to mycotoxins or the effect of mycotoxin on the growth of insects and the identification of the produced metabolites. Research on the breakdown/reduction of mycotoxins by edible insects is still in an early stage. However, more research needs to be conducted on the metabolism pathway for the degradation of mycotoxins by each species, the identification and isolation of enzymes involved in the metabolism, and the toxicity of the metabolites. Through a review of the current research status, the following research gaps have been identified, and a focus of future research could be derived from there.
Mycotoxins in insect feed need to be further determined. Food waste could possibly be used as insect feed in order to reduce levels of food waste. Therefore, the mycotoxins in food waste need to be identified and quantified. This knowledge will identify which mycotoxins and at what concentration should be fed to edible insects to study their metabolism and their effect on insects. Currently the mycotoxins that have been studied include only aflatoxins, fumonisins, zearalenones, deoxynivalenol, and ochratoxins. There are possibly more types of mycotoxins in food waste and agricultural products.Further understanding of the effect of each mycotoxin on each edible insect needs to be provided. A larger range of mycotoxin concentrations in experimental insect feeding trials should be included to determine the toxicity threshold of each different insect. This knowledge could be an essential reference for determining the inclusion level of mycotoxin-contaminated food waste in an insect diet.The metabolites of each mycotoxin by each insect need to be identified. Information on this is currently very limited. Additional scientific research needs to be conducted for this purpose.Investigations on the toxicity of each metabolite must be carried out for food safety reasons.The metabolic pathway of each mycotoxin in each insect needs to be explored, and the enzymes involved in the breakdown of each mycotoxin could be determined and purified. These enzymes might be used in the food and feed industry for mitigating the mycotoxin problem.

## Figures and Tables

**Table 1 toxins-14-00217-t001:** Summary of mycotoxin breakdown by insect metabolism.

Insect	Mycotoxin (s)	Metabolites	Enzymes	Detoxification Mechanism	References
*Apis mellifera* (Honeybee, human food)	AFB1 *^a^*	--	P450 *^b^*	--	[18,19,20]
*Helicoverpa zea* (Corn earworm, feed for chickens)	AFB1	AFP1 *^c^*	P450 CYP321A1	Metabolic activity enhanced by phytochemicals (xanthotoxin, coumarin)	[19,21,22]
*Tenebrio molitor* (Yellow mealworm, human food and animal feed)	ZEN *^d^*	α-, β-ZEL *^e^*	--	Metabolic degradation and rapid excretion No sulfonation occurs	[7]
	AFB1	AFM1 *^f^*	P450	Gut contents play a role in metabolic degradation Rapid excretion	[23]
	DON *^g^*	3-acetyl-DON NIV *^h^*	Phase II enzymes SOD *^i^* GST *^j^*	Acetylation by midgut and gut bacterial enzymes Detoxification of reactive oxygen species Rapid excretion	[4,24,25]
	FB1 *^k^*	--	--	Immobilization within fat storages, excretion, or metabolic degradation	[26]
*Hermetia illucens* (Black soldier fly, animal feed)	AFB1	AFL	P450	Metabolic degradation Rapid excretion	[2,23]
	DON	--	--	Metabolic degradation by gut bacteria Rapid excretion	[2,5]
	OTA *^l^*	--	--	Metabolic degradation and excretion	[2]
	ZEN	α-, β-ZEL	--	Over 50% of ZEN converted to its metabolites Excretion	[2]
*Alphitobius diaperinus* (Lesser mealworm, animal feed)	AFB1	AFL *^m^* AFM1	--	Limited AFB1 metabolism Rapid excretion	[2]
	DON	--	--	Metabolic degradation and rapid excretion	
	OTA	--	--	Excretion	
	ZEN	α-, β-ZEL	--	Limited ZEN metabolism Rapid excretion	

*^a^* AFB1—aflatoxin B1; *^b^* P450—cytochrome P450 monooxygenase; *^c^* AFP1—aflatoxin P1; *^d^* ZEN—zearalenone; *^e^* ZEL—zearalenol; *^f^* AFM1—aflatoxin M1; *^g^* DON—deoxynivalenol; *^h^* NIV—nivalenol; *^i^* SOD—superoxide dismutase; *^j^* GST—glutathione-S-transferase; *^k^* FB1—fumonisin B1; *^l^* OTA—ochratoxin A; *^m^* AFL—aflatoxicol.

## Data Availability

Not applicable.

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
