# Peer review of "Mycotoxin Metabolism by Edible Insects"

_toxins, 2022, doi:10.3390/toxins14030217_

Round 1

Reviewer 1 Report

Mycotoxin contamination is one of the major issues causing economic losses in human food and farm animal production. The toxicity of various mycotoxins has been investigated for years, solving the issues of mycotoxins is a challenge to reachers worldwide. Although lots of work has been done, studies to eliminate the mycotoxins from contaminated feed have not been successful. The current review manuscript summarized a group of great research using insects to make better use of the contaminated materials. The direction of such research may have a huge economic impact if successful strategies can be developed.

I enjoyed reading the review paper, and it is very comprehensive in summarizing both background information and the latest progress. The manuscript is well prepared, written English is perfect, and the structure of the manuscript is also very clear. I think the review manuscript is ready to be published as it is.

Author Response

Dear reviewer,

Thanks for your comments!

Reviewer 2 Report

Dear Authors, 

The presented work "Mycotoxin Metabolism by Edible Insects" raises an important and interesting issue. Limitation of animal (mammalian) protein production capacity and new consumer preferences oblige scientists to search for healthy alternative sources of protein. One such solution is the use of insect protein in human diets and animal feeds. 
However, the work I reviewed is not scientifically mature. Why? A review article should also add value to science. The work is currently chaotic and does not point out specific gaps in the research. I suggest indicating in Table 1 all edible insect species currently identified in the literature and in this table indicate the place of use (food, feed for which animal species) and if the metabolism of mycotoxins. This presentation of the problem will make it possible to indicate the gap in the current state of knowledge. I also suggest presenting the harmfulness of given mycotoxins to insects in one place and a complete listing of their occurrence in agricultural products in another place. 
The conclusions of such an article should also be more thoughtful/insightful. Currently they are more of a summary.

Author Response

Thanks to your comments, responses to each point is provided  below in italic. 

The presented work "Mycotoxin Metabolism by Edible Insects" raises an important and interesting issue. Limitation of animal (mammalian) protein production capacity and new consumer preferences oblige scientists to search for healthy alternative sources of protein. One such solution is the use of insect protein in human diets and animal feeds. 
However, the work I reviewed is not scientifically mature. Why? A review article should also add value to science. The work is currently chaotic and does not point out specific gaps in the research. I suggest indicating in Table 1 all edible insect species currently identified in the literature and in this table indicate the place of use (food, feed for which animal species) and if the metabolism of mycotoxins. This presentation of the problem will make it possible to indicate the gap in the current state of knowledge.

Response: This table in fact included the most commonly used edible insects. We added the place of use in the revision.

 I also suggest presenting the harmfulness of given mycotoxins to insects in one place and a complete listing of their occurrence in agricultural products in another place. 

Response: As indicated in the this review, all insects have a certain level of tolerance to mycotoxins. The harmfulness of mycotoxins to insects are dependent on the concentration of the mycotoxins. As this review is focused on the metabolism of mycotoxins and research on the toxicity of mycotoxins to insects is very limited, we didn’t list the harmfulness of mycotoxins to insects.

The conclusions of such an article should also be more thoughtful/insightful. Currently they are more of a summary.

Response: We have added our insights in the new version in the conclusion.

Reviewer 3 Report

In summary, in order to understand metabolic capabilities and potential toxic effects on organisms, studies have been conducted in which edible insects are exposed to naturally occurring and naturally occurring levels of mycotoxins. Mycotoxin concentrations were analyzed and subsequent metabolites from insects after short or prolonged periods of exposure. These aspects have led to an understanding of how these microorganisms manage toxins. The authors paid special attention to the publications of the last 20 years focused on the detoxification of mycotoxins by edible insects, in order to provide an overview of scientific research in this field and to identify gaps for future research.

The bibliography is recent and adequate, representing a good critical analysis of the research field. The analytical methods used are well selected and allow the realization of both co-assistance and corroboration systems between experimental techniques. The graphics are representative and well done. The conclusions sound well written. The paper can be published in its current form.      

Author Response

Thanks for your positive comments!

Reviewer 4 Report

The topic of this review is interesting and intends to highlight the scientific evidence supporting the problematics of mycotoxins, specifically centring on the role of edible insects. This is very pertinent and directly links with food safety as well as economic losses due to foods that have to be discarded because of contamination with mycotoxins.

However, the present review dos not quite meet the criteria to be considered for publication.

The introductory section is too brief and lacks some contextualization and further justification to introduce the topic of the review.

Although being a review, this should include a methodology section explaining how the review was conducted. If it followed any systematic review approach or what kind of criteria were used to include or exclude the studies.

The development and the conclusions are weak and the studies focused are limited. The number of references is more adequate for a research article than a review. The approach is not deep enough and the final observations do not quite answer the possible questions that should be highlighted as objectives.

From my point of view this work does not meet the necessary quality criteria to be accepted for publication.

Author Response

Thanks for your comments, responses to each point are provided below in italic. 

The topic of this review is interesting and intends to highlight the scientific evidence supporting the problematics of mycotoxins, specifically centring on the role of edible insects. This is very pertinent and directly links with food safety as well as economic losses due to foods that have to be discarded because of contamination with mycotoxins.

However, the present review dos not quite meet the criteria to be considered for publication.

Response: We have revised the manuscript based on the comments from reviewers and we are sure this manuscript provided essential knowledge and insight on the metabolism of mycotoxins by edible insects. We are confident that it meets the criteria for publication. 

The introductory section is too brief and lacks some contextualization and further justification to introduce the topic of the review.

Response: More information has been added in the introduction to provide contextualization and justification to introduce  the topic of the review.

Although being a review, this should include a methodology section explaining how the review was conducted. If it followed any systematic review approach or what kind of criteria were used to include or exclude the studies.

Response: The following information on how the review was conducted was added in the manuscript. “With the purpose to provide the general situation of scientific research on this area and identify gaps for future research, scientific literatures on insects and mycotoxins that published in the last 20 years were searched in Agricola databases, CAB databases and Scopus. The literatures on edible insects were first summarized and then the metabolism of insects were summarized based on the group of mycotoxins including aflatoxins, fumonisins, zearalenones, vomitoxin or deoxynivalenol, and ochratoxins. Future directions and recommendations on this topic were provided in the conclusion part.”

The development and the conclusions are weak and the studies focused are limited. The number of references is more adequate for a research article than a review. The approach is not deep enough and the final observations do not quite answer the possible questions that should be highlighted as objectives.

Response: More information on insight for future work has been provided in the conclusion part as below.  Research on “ insects and mycotoxins” are on the early stage and  we used several database to search papers on this topic and those are all we found. We hope our review could attract more research work on this important area.

 “Through the review of current research status, the following research gaps have been identified and focus of future research could be derived from there.

  1. Mycotoxins in insect feed need to be further determined. Food waste could be possibly used as insect feed for reducing food waste. Therefore the mycotoxins in those food waste need to be identified and quantified. This knowledge will identify which mycotoxins and at what concentration should be fed to edible insects to study their metabolism and their effect on insects. Currently the mycotoxins that have been studied only include aflatoxins, fumonisins, zearalenones, deoxynivalenol, and ochratoxins. There are possibly more types of mycotoxins in food waste and agricultural products.
  2. Further understanding of the effect of each mycotoxin on each edible insect needs to provide. Larger range of mycotoxin concentrations in the experiment insect feeding trial should be included to determine the toxicity threshold to each different insect. This knowledge could be essential reference for determining the inclusion level of mycotoxin contaminated food waste in insect diet.
  3. Metabolites of each mycotoxin by each insects needs to identified. Information on this is still very limited currently. Additional scientific research needs to be conducted for this purpose.
  4. Investigations on the toxicity of each metabolites must be carried out for food safety reason.
  5. Metabolism pathway of each mycotoxin in each insects needs to be explored, and enzymes involved in the breakdown of each mycotoxin could be determined and purified. Those enzymes might used in food and feed industry for mitigating mycotoxin problem.

From my point of view this work does not meet the necessary quality criteria to be accepted for publication.

Response: I am sorry to hear this, but we are sure this review provided essential knowledge and insights on the research area of mycotoxin metabolism by edible insects. We are confident that it meets the criteria to be accepted for publication. 

Reviewer 5 Report

The manuscript titled “Mycotoxin Metabolism by Edible Insects” is an important article covering different aspects of edible insects for alternative sources of animal protein.

The submitted article is well organized and well written. The authors have covered all the aspects of mycotoxins by edible insects industriously and thoroughly.  

There are, however, some issues with the manuscript which should be addressed for its improvement.

The abstract should be revised to make it robust. It should succinctly cover all the aspects of the mycotoxins by insects. More information regarding mycotoxins should be added.  

Keywords correspond to the aim.

The introduction is specific and focused on.

The conclusion needs revision and the authors should give some future directions and recommendations.

The standard of English is acceptable however, some syntactical errors were spotted.

Some of the references do not follow the format of the journal.

Many stylistic errors have also been spotted.

Author Response

Thanks to your comments! The response to each points is provided below in italic.

The manuscript titled “Mycotoxin Metabolism by Edible Insects” is an important article covering different aspects of edible insects for alternative sources of animal protein.

The submitted article is well organized and well written. The authors have covered all the aspects of mycotoxins by edible insects industriously and thoroughly.

Response: Thanks for your positive comments.

There are, however, some issues with the manuscript which should be addressed for its improvement.

Response: We have revised the manuscript based on your comments below.

The abstract should be revised to make it robust. It should succinctly cover all the aspects of the mycotoxins by insects. More information regarding mycotoxins should be added.

Response: More information regarding mycotoxin has been added. More specifically, the following information was added in the abstract. “ Aflatoxin metabolism is proved through the production of hydroxylated metabolites by NADPH-dependent reductases and hydroxylases by different insects. ZEN can be metabolized into a- and b- zearalenol. Three DON metabolites 3-, 15-acetyl-DON, and DON-3-glucoside have been identified in the insect DON metabolites. Unfortunately, the resulting metabolites, involved enzymes and detoxification mechanisms of OTAs and fumonisins within insects have yet to be identified.”

Keywords correspond to the aim.

Response: Thanks.

The introduction is specific and focused on.

Response: Thanks.

The conclusion needs revision and the authors should give some future directions and recommendations.

Response: Future directions and recommendations have been added in the new version as below. “ Through the review of current research status, the following research gaps have been identified and focus of future research could be derived from there.

  1. Mycotoxins in insect feed need to be further determined. Food waste could be possibly used as insect feed for reducing food waste. Therefore the mycotoxins in those food waste need to be identified and quantified. This knowledge will identify which mycotoxins and at what concentration should be fed to edible insects to study their metabolism and their effect on insects. Currently the mycotoxins that have been studied only include aflatoxins, fumonisins, zearalenones, deoxynivalenol, and ochratoxins. There are possibly more types of mycotoxins in food waste and agricultural products.
  2. Further understanding of the effect of each mycotoxin on each edible insect needs to provide. Larger range of mycotoxin concentrations in the experiment insect feeding trial should be included to determine the toxicity threshold to each different insect. This knowledge could be essential reference for determining the inclusion level of mycotoxin contaminated food waste in insect diet.
  3. Metabolites of each mycotoxin by each insects needs to identified. Information on this is still very limited currently. Additional scientific research needs to be conducted for this purpose.
  4. Investigations on the toxicity of each metabolites must be carried out for food safety reason.
  5. Metabolism pathway of each mycotoxin in each insects needs to be explored, and enzymes involved in the breakdown of each mycotoxin could be determined and purified. Those enzymes might used in food and feed industry for mitigating mycotoxin problem. “

The standard of English is acceptable however, some syntactical errors were spotted.

Response: We have double checked and corrected. Thanks.

Some of the references do not follow the format of the journal.

Response: We have double checked and corrected. Thanks.

Many stylistic errors have also been spotted.

Response: We have double checked and corrected. Thanks.

Round 2

Reviewer 2 Report

Dear Authors, 

the corrected article looks much better. However, you have not indicated in the table the use for all species. Furthermore, if you change the scope of a column, you should change its heading. In its current form, the reader is misled. In this table, does the use of insects fit in with the cited publications? It should be indicated for which specific animals insects are food, not only "animal food".

Author Response

Dear reviewer,

Thank you very much for your comments, they are very good points. Please find the response to your comments below.

the corrected article looks much better.

Response: Thanks.

However, you have not indicated in the table the use for all species.

Response: We deleted the one that we can not find its use for animal feed or human food.

 Furthermore, if you change the scope of a column, you should change its heading. In its current form, the reader is misled.

Response: We decided not to change the scope of the column. So the heading is kept.

In this table, does the use of insects fit in with the cited publications?

Response: We added those references for those species whose use as feed or food were not in the already cited publication. But most of their use as animal feed or human food have been stated in the already cited publications.

It should be indicated for which specific animals insects are food, not only "animal food".

Response: For the “animal food”, it indicates it is good as a protein source and could be fed to most animals. Only “Helicoverpa zea (Corn earworm, feed for chickens)” was specified for chicken because it is not used as commonly as the other insects as animal feed, and so far only chicken has been found to eat this species.

Reviewer 4 Report

The authors have improved the manuscript and now I beleve that it can in fact be accepted.

Author Response

Thanks for your positive comments!